



**Interferences on Aerosol Acidity Quantification due to Gas-phase Ammonia Uptake onto**
**Acidic Sulfate Filter Samples**
Benjamin A. Nault[1,2], Pedro Campuzano-Jost[1,2], Douglas A. Day[1,2], Hongyu Guo[1,2], Duseong S.
Jo[1,2,*], Anne V. Handschy[1,2], Demetrios Pagonis[1,2], Jason C. Schroder[1,2,**], Melinda K.
Schueneman[1,2], Michael J. Cubison[3], Jack E. Dibb[4], Alma Hodzic[5], Weiwei Hu[6], Brett B. Palm[7],
Jose L. Jimenez[1,2]
1. Department of Chemistry, University of of Colorado, Boulder, CO, USA
2. Cooperative Institute for Research in Environmental Sciences, University of Colorado,
Boulder, CO, USA
3. TOFWERK AG, Boulder, CO USA
4. Earth Systems Research Center, Institute for the Study of Earth, Oceans, and Space,
University of New Hampshire, Durham, NH, USA
5. Atmospheric Chemistry Observations and Modeling Laboratory, National Center for
Atmospheric Research, Boulder, CO, USA
6. State Key Laboratory at Organic Geochemistry, Guangzhou, Institute of Geochemistry,
Chinese Academy of Sciences, Guangzhou, China
7. Department of Atmospheric Sciences, University of Washington, Seattle, WA, USA
* Now at: Advanced Study Program, National Center for Atmospheric Research, Boulder, CO
USA
** Now at: Colorado Department of Public Health and Environment, Denver, CO, USA
Correspondence: Jose L. Jimenez (jose.jimenez@colorado.edu)



**Abstract**

Measurements of the mass concentration and chemical speciation of aerosols are important to investigate their chemical and physical processing from near emission sources to the most remote regions of the atmosphere. A common method to analyze aerosols is to collect them onto filters and to analyze filters off-line; however, biases in some chemical components are possible due to changes in the accumulated particles during the handling of the samples. Any biases would impact the measured chemical composition, which in turn affects our understanding of numerous physico-chemical processes and aerosol radiative properties. We show, using filters collected onboard the NASA DC-8 and NSF C-130 during six different aircraft campaigns, a consistent, substantial difference in ammonium mass concentration and ammonium-to-anion ratios, when comparing the aerosols collected on filters versus the Aerodyne Aerosol Mass Spectrometer (AMS). Another *on-line* measurement is consistent with the AMS in showing that the aerosol has lower ammonium-to-anion ratios than obtained by the filters. Using a gas uptake model with literature values for accommodation coefficients, we show that for ambient ammonia mixing ratios greater than 10 ppbv, the time scale for ammonia reacting with acidic aerosol on filter substrates is less than 30 s (typical filter handling time in the aircraft) for typical aerosol volume distributions. Measurements of gas-phase ammonia inside the cabin of the DC-8 show ammonia mixing ratios of 45±20 ppbv, consistent with mixing ratios observed in other indoor environments. This analysis enables guidelines for filter handling to reduce ammonia uptake. Finally, a more meaningful limit-of-detection for filters that either do not have an ammonia scrubber and/or are handled in the presence of human emissions is ~0.2 μg m$^{-3}$ ammonium, which is substantially higher than the limit-of-detection of the ion chromatography.





**Introduction**

Particulate matter (PM), or aerosol, impacts human health, ecosystem health, visibility,
climate, cloud formation and lifetime, and atmospheric chemistry (Meskhidze et al., 2003;
Abbatt et al., 2006; Seinfeld. and Pandis, 2006; Jimenez et al., 2009; Myhre et al., 2013; Cohen
et al., 2017; Hodzic and Duvel, 2018; Heald and Kroll, 2020; Pye et al., 2020). Quantitative
measurements of the chemical composition and aerosol mass concentration are necessary to
understand these impacts and to constrain and improve chemical transport models (CTMs). The
inorganic portion of aerosol, which includes both volatile (e.g., nitrate, ammonium) and
non-volatile (e.g., calcium, sodium) species, controls many of these impacts through the
regulation of charge balance, aerosol pH, and aerosol liquid water concentration (Guo et al.,
2015, 2018; Hennigan et al., 2015; Nguyen et al., 2016; Pye et al., 2020). Further, the inorganic
portion of aerosol is an important fraction of the aerosol budget, both in polluted cities (e.g.,
Jimenez et al., 2009; Song et al., 2018), and remote regions (e.g., Hodzic et al., 2020), and the
chemistry controlling the inorganic portion of the aerosol is still not well known (e.g., Liu et al.,

2020).

There are numerous methods to quantify the inorganic aerosol composition and mass
concentration, including by mass spectrometry (DeCarlo et al., 2006; Canagaratna et al., 2007;
Pratt and Prather, 2010; Froyd et al., 2019), *on-line* ion chromatography (Talbot et al., 1997;
Weber et al., 2001; Nie et al., 2010), and collection onto filters to be extracted and measured
off-line by ion chromatography (Malm et al., 1994; Dibb et al., 2002, 2003; Coury and Dillner,
2009; Watson et al., 2009). Each method has different advantages and disadvantages (e.g., time
resolution, sample preparation, range of species identified, cost, and personnel needs). These



results, in turn, have been used to inform and improve the results of CTMs, influencing our
understanding in processes such as the direct radiative effect (Wang et al., 2008b), transport of
ammonia in deep convection (Ge et al., 2018), aerosol pH (Pye et al., 2020; Zakoura et al., 2020)
and subsequent chemistry, and precursor emissions (Henze et al., 2009; Heald et al., 2012;
Walker et al., 2012; Mezuman et al., 2016).
Filter measurements have been shown to be most prone to artifacts during sample
collection, handling, storage of the filter, or extraction of the aerosol from the filter prior to
analysis. These artifacts include evaporation of volatile compounds such as organics (Watson et
al., 2009; Chow et al., 2010; Cheng and He, 2015) and ammonium nitrate (Hering and Cass,
1999; Chow et al., 2005; Nie et al., 2010; Liu et al., 2014, 2015; Heim et al., 2020), as well as
chemical reactions of gas-phase species with the accumulated particles (e.g., Schauer et al.,
2003; Dzepina et al., 2007). Further, early research indicated potential artifacts from gas-phase
ammonia uptake onto acidic aerosol collected onto filters, leading to a positive bias for
particulate ammonium (Klockow et al., 1979; Hayes et al., 1980; Koutrakis et al., 1988). This led
to debates about whether aerosol in the lower stratosphere was sulfuric acid or ammonium
sulfate (Hayes et al., 1980); however, after improved filter handling practices and *on-line*
measurements (i.e., mass spectrometry), it has been generally well accepted that the sulfate in the
stratosphere is mainly sulfuric acid (Murphy et al., 2014).
This artifact may impact aerosol collected in remote locations (e.g., the lower
stratosphere, but also the free troposphere over the Pacific Ocean basin). Comparisons for a
major cation, ammonium, in a similar location (middle of the Pacific Ocean) have shown very
different results (Dibb et al., 2003; Paulot et al., 2015). This, in turn, affects the observed charge





balance of cations (sulfate and nitrate) with ammonium, which can indicate different aerosol
phase state (Colberg et al., 2003; Wang et al., 2008a) and aerosol pH (Pye et al., 2020), leading
to potentially important chemical and physical differences between the real state of the particles
and that concluded from the measurements. An example of the differences in observed charge
balance of ammonium to sulfate for different studies of the same remote Pacific Ocean region is
highlighted in Fig. 1. This difference leads to the inorganic portion of the aerosol potentially
being solid (filters) and hence good ice-nucleating particles (Abbatt et al., 2006), versus it being
liquid (*on-line* measurements), leading to important differences in the calculated radiative
balance. It should be noted that other measurements (both filter and *on-line*) in a similar location
from another study (bar at surface (Paulot et al., 2015)) are more in-line with the *on-line*
observations. A large decrease in the ambient ammonia mixing ratio is required to change from
ammonium sulfate-like aerosols to sulfuric acid-like aerosols between the years, contradictory to
the increasing trends of ammonia globally (Warner et al., 2016, 2017; Weber et al., 2016; Liu et
al., 2019; Tao and Murphy, 2019). Further, oceanic emissions of ammonia are not high enough to
lead to full charge neutralization of sulfate, since these emissions are approximately an order of
magnitude less than those of sulfate precursors (Faloona, 2009; Paulot et al., 2015). A debate
about the acidity and potential impact of ammonia-uptake artifacts on acidic filters for remote
locations has not occurred as it did for stratospheric observations.

Previous laboratory studies have suggested that exposure of acidic aerosol, both

suspended in air in a flow tube or on a filter, to gas-phase ammonia will lead to formation of
ammonium salts in short time ($\leq 10$ s) (Klockow et al., 1979; Huntzicker et al., 1980); however,
it has not been investigated if this time frame applies for acidic aerosol collected on filters



handled in a typical indoor environment. Though human emissions of ammonia are variable and
depend on various factors (e.g., temperature, clothing, etc.) (Li et al., 2020), the emissions of
ammonia, specifically from perspiration but also from breath, can lead to high, accumulated
mixing ratios of ammonia indoor (e.g., Ampollini et al., 2019; Finewax et al., 2020) and
references therein), depending on the ventilation rate. The mixing ratios of ammonia can be
factor of 2 to 2000 higher indoor versus outdoor. This higher mixing ratio of ammonia leads to
similarly high mixing ratios used in prior studies to lead to partially to fully neutralize sulfuric
acid (Klockow et al., 1979; Huntzicker et al., 1980; Daumer et al., 1992; Liggio et al., 2011).

Here, we investigate whether previously observed laboratory observations of ammonium

uptake to acidic particulate lead to the large differences in ammonium, both in mass
concentration and in ammonium-to-sulfate ratios or ammonium-to-anion ratios, between *in-situ*
measurements and *off-line* filter measurement during five NASA and one NSF airborne
campaigns that sampled air over remote continental and oceanic regions. An uptake model for
gas-phase ammonia interacting with acidic PM on a filter along with constraints from
observations of gas-phase ammonia in the cabin of the airplane are used to further probe the
reason behind the differences between the *in-situ* and *off-line* measurements of ammonium. The
results provide insight into how to interpret prior aircraft measurements and other filter based
measurements where the filters were handled in environments (i.e., indoors), where rapid uptake
of ammonia to acidic PM will occur.

**2. Methods**
**2.1 Aircraft Campaigns**



Five different NASA aircraft campaigns on-board the DC-8 research aircraft and one

NSF aircraft campaign on-board the C-130 research aircraft are used in this study. As described
below, though the campaigns were sampling ambient (outside) air in various locations around the
world, the filters were handled and exposed to both aircraft cabin air and indoor temporary
laboratory air, where between 20 and 40 people were operating instruments. The campaigns
include the Arctic Research of the Composition of the Troposphere from Aircraft and Satellites
(ARCTAS) -A (April 2008) and -B (June – July 2008) campaigns (Jacob et al., 2010), the
Studies of Emissions and Atmospheric Composition, Clouds, and Climate Coupling by Regional
Surveys (SEAC$^4$RS, August – September 2013) campaign (Toon et al., 2016), the Wintertime
INvestigation of Transport, Emissions, and Reactivity (WINTER, February – March 2015)
(Schroder et al., 2018), and the Atmospheric Tomography (ATom) -1 (July – August 2016) and
-2 (January – February 2017) campaigns (Hodzic et al., 2020). ARCTAS-A was based in
Fairbanks, Alaska, Thule, Greenland, and Iqaluit, Nunavut, and sampled the Arctic Ocean and
Arctic regions of Alaska, Canada, and Greenland; while, ARCTAS-B was based in Cold Lake,
Alberta, Canada, and sampled the boreal Canadian forest, including wildfire smoke. SEAC$^4$RS
was based in Houston, Texas, and sampled biomass burning from western forest fires and
agricultural burns along the Mississippi River and the Southern United States, isoprene
chemistry over Southern United States and midwestern deciduous forests, and deep convection
associated with isolated thunderstorms, the North American Monsoon, and tropical depressions.
Finally, ATom-1 and -2 sampled the remote atmosphere over the Arctic, Pacific, Southern, and
Atlantic Oceans during the Northern (Southern) Hemispheric summer (winter) and winter
(summer).



For ARCTAS-A, -B, and SEAC⁴RS, the general sampling scheme was regional, sampling
large regions at level flight tracks. ATom-1 and -2, being global in nature, only sampled at level
legs for short durations (5 – 15 min) at low (~300 m) and high (10 – 12 km) altitude, and did not
measure at level altitudes between the low and high altitude. Due to the sampling time of the
filters (see Sect. 2.2.2), the entirety of the ascent and descent time was needed for one filter
sample. Therefore, all data during the ascents and descents have not been considered in this
study to minimize any issues due to the mixing of aerosols of different compositions and
acidities.

**2.2 Aerosol Measurements**
**2.2.1 Aerosol Mass Spectrometer**
An Aerodyne High-Resolution Time-of-Flight Aerosol Mass Spectrometer, flown by the
University of Colorado-Boulder (CU for short), was flown during the five campaigns used here.
The general features of the AMS have been described in prior studies (DeCarlo et al., 2006;
Canagaratna et al., 2007), and the specifics of the CU AMS for each campaign has been
described elsewhere (Cubison et al., 2011; Liu et al., 2017; Nault et al., 2018; Schroder et al.,
2018; Guo et al., 2020; Hodzic et al., 2020). In brief, the AMS measured the mass concentration
of non-refractory species in $PM_1$ (PM with an aerodynamic diameter less than 1 μm, see Guo et
al. (2020) for details). Ambient air was sampled by drawing air through an NCAR
High-Performance Instrumental Platform for Environmental Modular Inlet (HIMIL; Stith et al.
(2009)) at a constant standard flow rate of 9 L min⁻¹ (T = 273.15 K and P = 1013 hPa). No active
drying of the sampling flow was used to minimize artifacts for semi-volatile species, but the



temperature differential between ambient and cabin typically ensured the relative humidity (RH)
inside the sampling line less than 40% (e.g., Nault et al., 2018). An exception to this was during
ATom-1 and -2, where the cabin temperature, along with the high RH in tropics, led to higher RH
in the sample lines in a few instances in the boundary layer, which was accounted for in the final
mass concentrations (Guo et al., 2020). The air sample was introduced into the AMS via an
aerodynamic focusing lens (Zhang et al., 2002, 2004), which was operated at 2.00 hPa (1.50
Torr), via a pressure-controlled inlet, which was operated at various pressures (94-325 Torr)
(Bahreini et al., 2008), depending on the ceiling of the campaign and lens transmission
calibrations (Hu et al., 2017b; Nault et al., 2018). The aerosol, once focused, was introduced into
a detection chamber after three differential pumping stages. The aerosol impacted on an inverted
cone porous tungsten "standard" vaporizer under high vacuum, which was held at ~600°C. Upon
impaction, the non-refractory portion of the aerosol (organic, ammonium, nitrate, sulfate, and
chloride) were flash-vaporized, and the vapors were ionized by 70 eV electron ionization. The
ions were then extracted and analyzed with a H-TOF time-of-flight mass spectrometer (Tofwerk
AG). The AMS was operated in the "V-mode" ion path (DeCarlo et al., 2006), with spectral
resolution ($m/\Delta m$) of 2500 at $m/z$ 44 and 2800 at $m/z$ 184. The collection efficiency (CE) for
AMS was estimated with the parameterization of Middlebrook et al. (2012), which has been
shown to perform well for ambient aerosols (Hu et al., 2017a, 2020). The AMS nominally
samples aerosol with vacuum aerodynamic diameter between 40 nm and 1400 nm, which was
calibrated for in SEAC[4]RS, ATom-1, and -2 (Liu et al., 2017; Guo et al., 2020). Software
packages Squirrel and PIKA under  Igor Pro 7 (WaveMetrics, Lake Oswego, OR) (DeCarlo et
al., 2006; Sueper, 2018) were used to analyze all AMS data.



A cryogenic pump, to reduce background of ammonium and organics (Nault et al., 2018;

Schroder et al., 2018), was flown on the AMS for SEAC[4]RS, ATom-1, and -2; but not for
ARCTAS-A and -B. The cryogenic pump lowers the temperature of a copper cylinder
surrounding the vaporizer to ~90 K. This freezes out the background gases and ensures low
detection limits from the beginning of the flight, which is critical since aircraft instruments can
typically not be pumped continuously and hence suffer from high backgrounds at switch-on. The
$2\sigma$ accuracy for the AMS for inorganic aerosol is estimated to be 35% (Bahreini et al., 2009; Guo
et al., 2020).

**2.2.2 Aerosol Filters**

Fast collection of aerosol particles onto filters during airborne sampling, via the

University of New Hampshire Soluble Acidic Gases and Aerosol (SAGA) technique, has been
described elsewhere (Dibb et al., 2002, 2003), and was flown during the five campaigns
investigated here. Briefly, air is sampled into the airplane via a curved leading edge nozzle (Dibb
et al., 2002). The inlet is operated isokinetically during flight, and typically has a 50% collection
efficiency for aerosol with an aerodynamic diameter of 4.1 μm (Dibb et al., 2002; McNaughton
et al., 2007), with some altitude dependence (Guo et al., 2020). Aerosol was collected onto
Millipore Fluoropore Teflon filters (90 mm diameter with 1 μm pore size). Collection time was
dependent on altitude and estimated mass concentration, but generally 2 to 3 sm$^3$ (where sm$^3$ is
standard m$^{-3}$ at temperature = 273 K and pressure = 1013 hPa) volume of air is collected to
ensure detectable masses of species (Dibb et al., 2002). The filters were contained in a Delrin
holder during collection. After collection, the filters were transferred to a particle free



polyethylene "clean room" bag, which was filled with zero air, sealed, and stored over dry ice.
The samples from the filters were then extracted during non-flight days with 20 mL ultrapure
water and preserved with 100 μL chloroform. The preserved samples were sent to the University
of New Hampshire, to be analyzed by ion chromatography. The estimated limit of detection for
both sulfate and ammonium is 0.01 μg sm$^{-3}$ for all missions evaluated here (Dibb et al., 1999).

### 2.2.3 Other Aerosol Measurements

The NOAA Particle Analysis by Laser Mass Spectrometer (herein PALMS) was flown

during ATom-1 and -2. Details of the PALMS instrument configured for ATom-1 and -2 are
described in Froyd et al. (2019). Briefly, PALMS measures the chemical composition of single
aerosol particles via laser-ablation/ionization (Murphy and Thomson, 1995; Thomson et al.,
2000), where the ions are extracted and detected by a time of flight mass spectrometer. The
instrument measures particles between 100 nm and 4.8 μm (geometric diameter) (Froyd et al.,
2019). The measurement of PALMS used in this study is the "sulfate acidity indicator" (Froyd et
al., 2009). These authors reported that in the negative ion mode, there is a prominent peak at $m/z$
97, corresponding to $HSO_4^-$, and another peak at $m/z$ 195, corresponding to the cluster
$HSO_4^-(H_2SO_4)$. The first peak was independent of acidity; whereas, the second peak was
dependent on acidity. Froyd et al. (2009) calibrated the PALMS ratio of
$HSO_4^-(H_2SO_4)/(HSO_4^- + HSO_4^-(H_2SO_4))$ to Particle-into-Liquid Sampler (PILS) measurements to
achieve an estimate of ammonium balance.



244  Besides the chemical composition, the particle number and volume distributions are used

245 here. For SEAC[4]RS, the measurements have been described elsewhere (e.g., Liu et al., 2016).

246 The laser aerosol spectrometer (from TSI), which measured aerosol from geometric diameter 100

247 nm to 6.3 μm, is used here for volume distribution. For the ATom missions, the measurements

248 have been described elsewhere (Kupc et al., 2018; Williamson et al., 2018; Brock et al., 2019).

249 Briefly, the dry particle size distribution, from geometric diameter of 2.7 nm to 4.8 μm, were

250 measured by a series of optical particle spectrometers, including the Nucleation Model Aerosol

251 Size Spectrometer (3 nm to 60 nm,  custom built (Williamson et al., 2018)), an Ultra-High

252 Sensitivity Aerosol Spectrometer (60 nm to 1 μm) from Droplet Measurement Technologies

253 (Kupc et al., 2018)), and Laser Aerosol Spectrometer (120 nm to 4.8 μm) from TSI). These

254 measurements have been split in nucleation mode (3 to 12 nm), Aitken mode (12 to 60 nm),

255 accumulation mode (60 to 500 nm) and coarse mode (500 nm to 4.8 μm).

257 **2.3 Gas-Phase and Other Measurements**

258 **2.3.1 Ammonia Measurements**

259  Gas-phase ammonia was measured inside the cabin of the NASA DC-8 during the

260 FIREX-AQ campaign (Warneke et al., 2018), a subsequent DC-8 campaign which shared many

261 instrument installations and a similar level of aircraft personnel with the campaigns analyzed

262 here. The location of the instrument and where it sampled cabin ammonia (in relation to where

263 the SAGA filters are located) is shown in Fig. S1. Ammonia was measured by a Picarro G2103

264 Gas Concentration Analyzer (von Bobrutzki et al., 2010; Sun et al., 2015; Kamp et al., 2019).

265 The instrument is a continuous, cavity ring-down spectrometer. Cabin air is brought into a cavity





at low pressure (18.7 kPa, 140 Torr), where laser light is pulsed into the cavity. The light is
reflected by mirrors in the cavity, providing an effective path length of kilometers. A portion of
the light penetrates the mirrors, reaching the detectors, where the intensity of the light is
measured to determine the mixing ratio of ammonia from the time decay of the light intensity via
Beer-Lambert Law. The instrument measures the absorption of infrared light from 6548.5 to
6549.2 cm$^{-1}$ (Martin et al., 2016). Absorption of gas-phase water is also measured and corrected
for. This water vapor measurement is also used to calculate RH inside the cabin of the DC-8
(Filges et al., 2018). Data was logged at 1 Hz.

**2.3.2 Carbon Dioxide and Temperature Measurements**

Carbon dioxide inside the cabin of the NASA DC-8 during FIREX-AQ was measured by

a HOBO MX1102 Carbon Dioxide Data Logger (HOBO by Onset). It is a self-calibrating carbon
dioxide sensor with a range of 0 to 5,000 ppm carbon dioxide and an accuracy of ±50 ppm. A
non-dispersive infrared sensor is used to measure carbon dioxide. Data was acquired once every
10 s to once every 2 min. Besides carbon dioxide, RH and temperature are also recorded by the
instrument. Prior to each flight, the instrument was turned on and measured ambient carbon
dioxide, outside the cabin of the DC-8, to ensure the accuracy of the instrument compared to
ambient carbon dioxide measurements.

Ambient carbon dioxide during FIREX-AQ was measured by an updated version of the

instrument known as Atmospheric Vertical Observations of $CO_2$ in the Earth's Troposphere
(AVOCET) (Vay et al., 2003, 2011). The updated instrument used a modified LI-COR model
7000 non-dispersive infrared spectrometer and measured carbon dioxide at 5 Hz.



Temperature in the cabin was measured by a thermocouple (SEAC[4]RS) or thermistor
(ATom-1 and 2) located in the AMS rack or a Vaisala probe located at the front of the airplane
(ARCTAS-A, -B, and SEAC[4]RS).

**2.4 Theoretical Ammonia Flux Model**
To investigate the possibility that the ammonia mixing ratio in the cabin of the DC-8 is
high enough to be taken up by acidic PM on a filter during the short time the filter is exposed to
cabin air prior to final storage, a theoretical uptake model was constructed to estimate the time
scale for ammonia to interact with all the acidic particles (Seinfeld. and Pandis, 2006). The
equations used for the model can be found in the Supplemental Information (Sect. S2). The
model was initialized with a range of ammonia mixing ratios (1 to 200 ppb) and a range of PM
diameters (10 to 1000 nm). The calculations were conducted at 298 K, which is within ±10 K of
typical temperatures inside the cabin of the NASA DC-8 during the five campaigns (Fig. S2). An
accommodation coefficient of 1 for ammonia onto acidic PM was assumed (Hanson and
Kosciuch, 2003), with a density of 1.8 g cm$^{-3}$ for sulfuric acid (Rumble, 2019). For the mass
transfer calculations, the transition regime (between the free molecular and continuum regimes)
equations were used, using the Fuchs and Sutugin parameterization (Fuchs and Sutugin, 1971).
The model was used to estimate the ammonia molecular flux to acidic PM on the filter (Eq. S3).
Finally, the molecular flux was used to estimate the time it would take all the particles to be
partially neutralized by ammonia in the cabin (Eq. S4), though this may be a lower limit
(Robbins and Cadle, 1958; Daumer et al., 1992).





## 3. Results and Discussion

### 3.1 Comparison of On-Line and Off-Line Ion Balances across the Tropospheric Column

SAGA and AMS co-sampled aerosols during multiple aircraft campaigns. Nitrate quickly evaporates from aerosols as the aerosols are transported away from source regions and is typically small in the global troposphere (DeCarlo et al., 2008; Hennigan et al., 2008; Hodzic et al., 2020). Thus, in Fig. 2 the mass concentrations for the two most important submicron contributors to ammonium balance, ammonium and sulfate, are compared from the aircraft campaigns. The campaigns generally sampled remote air, either continental or oceanic, except for biomass burning sampled during ARCTAS-B and SEAC⁴RS and downwind of urban areas during WINTER. The measurements, for mass concentrations greater than 0.1 μg sm$^{-3}$, are generally within the combined uncertainties of the two instruments. Sulfate generally remains on the one-to-one line, even at low mass concentrations. However, ammonium shows a large divergence between the two measurements for mass concentrations less than 0.1 μg sm$^{-3}$ during all six aircraft campaigns. As shown in Fig. 2, the divergence in ammonium occurs well above the limit-of-detection for both instruments, namely ~4 ng sm$^{-3}$ for AMS for a 5-minute average (DeCarlo et al., 2006; Guo et al., 2020) and 10 ng sm$^{-3}$ for SAGA (Dibb et al., 1999), for both ammonium and sulfate.

This divergence in ammonium mass concentration is thus reflected in the ammonium balance, defined as the ratio of ammonium to sulfate plus nitrate, in moles (Fig. 3). For all campaigns, the two measurements show differences in ammonium balance, especially at higher altitudes, where the aerosols is distant from ammonia emissions (Dentener and Crutzen, 1994; Paulot et al., 2015), but sulfate production can continue due to vertical transport of precursors



such as $SO_2$. On average, the SAGA measurements indicate ammonium balance rarely below 0.5
throughout the troposphere; whereas, the AMS measurements indicate that ammonium balance
generally drops to below 0.2 for pressures less than 400 hPa. Fig. 2 and Fig. 3 indicate either
differences in the ammonium balance due to differences in aerosols population sampled, as
SAGA measures larger aerosols diameters than AMS (Guo et al., 2020), or potential artifacts
with one of the measurements.

Both the AMS and the filters sample most of the submicron aerosols (see Guo et al.

(2020) for details), but the filters also sample supermicron particles that the AMS does not.
Therefore it is possible in principle that the difference could be due to ammonium present in
supermicron particles. As discussed in Guo et al. (2020), nearly 100% of the measured volume
occurs for aerosols < 1 μm above the marine boundary layer, where the largest difference in
ammonium balance between the filters and AMS occurs (Fig. 3). Further, ammonium has been
observed to be a small fraction of the supermicron mass (Kline et al., 2004; Cozic et al., 2008;
Pratt and Prather, 2010), except for instances of continental fog (Yao and Zhang, 2012) and
Asian dust events (Heim et al., 2020). An upper estimate of supermicron ammonium can be
calculated  using results from prior studies (Kline et al., 2004; Cozic et al., 2008). In these prior
studies, ~90% of the ammonium was submicron. With the average ammonium observed during
ATom-1 and -2 (~10 to 50 ng sm$^{-3}$) (Hodzic et al., 2020), that would suggest an upper limit of ~1
to 5 ng sm$^{-3}$ ammonium in the supermicron aerosols. This upper estimate does not explain the
differences between AMS and filters during ATom-1 and -2 (Fig. S3), as the percent difference
increases with decreasing estimated supermicron ammonium volume. As the largest differences
between the AMS and filters occur well above the boundary layer (Fig. 3), away from



continental ammonia sources (Dentener and Crutzen, 1994) and Asian dust events, we conclude
that the sampling of supermicron aerosols by filters is not leading to the observed differences in
ammonium.

Prior studies by PALMS have shown aerosols observed for pressure < 400 hPa to be

acidic, depending on potential recent influence of boundary layer air via convection (Froyd et al.,
2009; Liao et al., 2015), similar to observations by other single particle mass spectrometers (Pratt
and Prather, 2010). Though not reaching similarly low $NH_4/(2\times SO_4)$ values as the AMS, the
PALMS acidity marker shows much lower values than were determined by the aerosols collected
on the filters (Fig. S4). Different reasons for PALMS not achieving as low values as AMS may
include differences in aerosols sizes sampled by PALMS versus AMS (Guo et al., 2020), and the
sensitivity of the acidity marker to laser power (Liao et al., 2015). Thus, two different *on-line*
measurements indicate that the ammonium balance is lower than the aerosols collected on filters,
suggesting potentially more acidic aerosols.

Differences in ammonium balance between AMS and SAGA are detectable for sulfate

mass concentrations $\leq 1$ μg sm$^{-3}$ (Fig. 4) for all six aircraft campaigns. As the sulfate mass
concentration decreases, the relative differences in ammonium, and thus ammonium balance,
increase. The large majority of the troposphere contains sulfate mass concentrations in which the
differences in ammonium are observed, highlighting the importance of this problem (Fig. 4a).
Thus, except for more polluted conditions (> 1 μg sm$^{-3}$ sulfate), which mainly occurs in
continental (Jimenez et al., 2009; Kim et al., 2015; Malm et al., 2017) and urban regions
(Jimenez et al., 2009; Hu et al., 2016; Kim et al., 2018; Nault et al., 2018), this bias between
filters and *on-line* measurements is critically important, especially since airborne measurements
are often the only meaningful observational constraints for remote regions. Thus, this analysis
suggest that for filters handled in indoor environments with large ammonia mixing ratios (see
below), a more meaningful ammonium limit-of-detection would be equivalent to 1 $\mu$g sm$^{-3}$
sulfate, which would be ~0.2 $\mu$g sm$^{-3}$ ammonium.

**3.2 Ammonia Levels on the NASA DC-8 Cabins**
Prior studies have suggested that various sources of ammonia could impact acidic filter
measurements (Klockow et al., 1979; Hayes et al., 1980; Koutrakis et al., 1988). Some of these
studies found that the materials of the containers where the filters are stored, unless thoroughly
cleaned and not stored around humans, are a source of ammonia gas that reacts with the sulfuric
acid on the filters to become ammonium, leading to ammonium bisulfate or ammonium sulfate
(Hayes et al., 1980). Further, handling of acidic filters in rooms with people or acidic aerosol in
the presence of human breath can also lead to near to complete neutralization of acidic aerosol
(Larson et al., 1977; Hayes et al., 1980; Clark et al., 1995). Finally, various studies have
suggested that the SAGA filters specifically may be impacted by various ammonia sources prior
to sampling with the ion chromatography (Dibb et al., 1999, 2000; Fisher et al., 2011).
During SAGA sampling, the filters with collected aerosol are moved from the sample
collector to a Teflon bag that is filled with clean air. During this step, the filter is exposed to the
cabin air of the DC-8 for ~10 s. As humans are a source of ammonia (Larson et al., 1977; Clark
et al., 1995; Sutton et al., 2000; Finewax et al., 2020; Li et al., 2020), this source sustains
significant ammonia concentrations in indoor environments, which could potentially bias the
filter measurements. *On-line* measurements would not be subject to this effect since the sampled



air is not exposed to cabin air before measurement. While inlet lines for *on-line* instruments
could in theory lead to some memory effects, there is no evidence of such effects in the data
(e.g., the response going from a large, neutralized plume into the acidic FT is nearly
instantaneous (Schroder et al., 2018)).
During a recent 2019 NASA DC-8 aircraft campaign, FIREX-AQ, ammonia was
measured on-board the DC-8 during several research flights. An example time series of cabin
ammonia, temperature, and RH is shown in Fig. 5. Prior to take-off, as scientists were slowly
boarding the airplane, the ammonia mixing ratio was low (< 20 ppbv) and similar to ambient
levels of ammonia outside the aircraft. As scientists started boarding before take-off, the
ammonia mixing ratio increased. Upon doors closing, the mixing ratio leveled off at ~40 ppbv.
After take-off, the mixing ratio remained ~40 ppbv, though there were changes related to
changes in cabin temperature and humidity, which would affect emission rates and also
adsorption of ammonia onto cabin surfaces (Sutton et al., 2000; Finewax et al., 2020; Li et al.,
2020) and movement of scientists throughout the cabin, which would affect emission rates and
their location.
The average (±1σ spread of the observations) and median ammonia in the cabin of the
DC-8 during FIREX-AQ was 45.4±19.9 and 41.9 ppbv (Fig. 6). There was a large positive tail in
ammonia mixing ratio, related to high temperatures (Fig. S5), which causes the scientists to
perspire more and release more ammonia (Sutton et al., 2000; Finewax et al., 2020; Li et al.,
2020). Compared to outdoor ammonia mixing ratios, ranging from urban to remote locations, the
ammonia in the cabin of the DC-8 is higher by a factor of 2 to 2000 (Fig. 6). On the other hand,





the ammonia measured in the cabin of the DC-8 is similar but towards the lower end of the
mixing ratios measured during various indoor studies (Table S1 for compilation of references).

The ammonia mixing ratios observed in the cabin were verified by investigating the cabin

air exchange rates (see SI Sect. S3). Using carbon dioxide measurements, the exchange rate in
the cabin was calculated to be 9.9 hr$^{-1}$ (Fig. S6), which is similar to literature values for the cabin
exchange rate of other passenger airliners (Hunt and Space, 1994; Hocking, 1998; Brundrett,
2001; National Research Council, 2002). This value is a factor of 2 to 5 higher than typical
exchange rates for commercial buildings (Hunt and Space, 1994; Pagonis et al., 2019), which
would suggest lower mixing ratios than observed in other indoor environments. Using this
exchange rate, and the literature total ammonia emission rates from humans (1940 μg hr$^{-1}$
person$^{-1}$ (Sutton et al., 2000)) and the average of ambient ammonia mixing ratios as an outdoor
background onto which the human emissions in the cabin are added (~4.4 ppbv, Fig. 6), the
ammonia mixing ratio in the cabin of the DC-8 was estimated to be 43.4 ppbv, which is within
the uncertainty of the average ammonia (45.4±19.9 ppbv) inside the cabin of the DC-8. Thus, the
observed ammonia mixing ratios in the cabin of the DC-8 are consistent with the cabin air
exchange rates and literature human ammonia emissions. These mixing ratios are approximately
a factor of nine higher than in a typical laboratory environment (Fig. S7), as there are fewer
people (1 to 4 versus 20 to 40), making the cabin of the DC-8 an extreme laboratory environment
for handling acidic filters. As shown in Fig. 6, ammonia mixing ratios in indoor environments
are high enough to change the thermodynamics of inorganic aerosol, leading to higher
ammonium balances (Weber et al., 2016). Thus, similar to the conclusions of other studies, the



cabin of the DC-8 is an important source of ammonia that could lead to biases with acidic

aerosols collected on filters.

### 3.3 Can Uptake of Cabin Ammonia Explain the Higher Ammonium Concentrations on Filters?

As shown in Fig. 6, the cabin of the DC-8 is an important source of ammonia from the

breathing and perspiring of scientists. Prior studies (Klockow et al., 1979; Huntzicker et al.,

1980; Daumer et al., 1992; Liggio et al., 2011) have shown in laboratory settings that 10 s is fast

enough to partially to fully neutralize sulfuric acid. Thus, here we investigate whether the time of

the filter handling of 10 s will lead to partial to full neutralization of sulfuric acid from cabin

ammonia, or whether this time is fast enough to limit exposure of the acidic filter to cabin

ammonia. Huntzicker et al. (1980) showed that for typical aerosol modal distributions (Fig. 7)

and cabin RH (Fig. S9), an initial pure sulfuric acid aerosol, suspended in a flow reactor, reaches

equal molar amounts of ammonium and sulfate (i.e., ammonium bisulfate) when exposed to 70

ppb ammonia in 10 s. This indicates the plausibility that acidic aerosol filters, which typically

have lower sulfate mass concentrations than Huntzicker et al. (1980) (~2 μg versus ~55 μg

sulfate equivalent on filters), would interact with cabin ammonia to form at least ammonium

bisulfate. Further, other studies found that in less than 10 s, sulfuric acid aerosol, suspended in a

flow reactor, at RH ≤ 45%, will completely react with gas-phase ammonia to form ammonium

sulfate (Robbins and Cadle, 1958; Daumer et al., 1992). The latter study used ammonia mixing

ratios similar to the amount observed in the cabin of the DC-8 (~30 ppbv); whereas, the former

study used excess ammonia (~9 ppmv).



First, the time of diffusion of ammonia gas from the surface to the interior of the filter
was investigated, as there is a potential for the PM to be embedded deep into the filter. Eq. 1
(Seinfeld. and Pandis, 2006):

$$\tau_{diffusion} = \frac{d_t^2}{8D_g}$$          Eq. 1

where  $d_t^2$ is the depth of the Teflon (~0.015 cm) and $D_g$ is the diffusion coefficient of ammonia in
air (0.228 cm$^2$ s$^{-1}$) (Spiller, 1989). Therefore, the estimated timescale for ammonia to diffuse
through the depth of the Teflon filter is ~1×10$^{-4}$ s, meaning that the surface of PM will always be
in contact with cabin-level mixing ratios of ammonia.
A theoretical uptake model for ammonia to acidic PM on filters was run for a range of
ammonia mixing ratios and PM diameters (Fig. 7). As shown in Fig. 7, only at the lowest
ammonia mixing ratios (< 10 ppbv), the flux of ammonia to acidic PM is slower (> 20 s) than the
typical filter handling time (~10 s) for typical aerosol diameters in the remote atmosphere. For
the conditions of the DC-8, similar to other indoor environments (> 20 ppbv ammonia, Fig. 6),
and ambient aerosol diameters in the accumulation mode that contains most ambient sulfate (Fig.
7), the amount of time needed for cabin ammonia to interact with acidic PM on filters to form
ammonium bisulfate is ≤ 10 s, similar to the results of Huntzicker et al. (1980). Also, studies
show that the kinetic limitation to form ammonium sulfate ($(NH_4)_2SO_4$) versus ammonium
bisulfate ($NH_4HSO_4$) is relatively low and can occur within the 10 s time frame (Robbins and
Cadle, 1958; Daumer et al., 1992). A laboratory setting with ~5 ppbv NH$_3$ would result in the
filters needing to be exposed to laboratory air for at least 40 s to form ammonium bisulfate (Fig.
S8) versus the 3 to 10 s for conditions in the cabin of the DC-8 (Fig. 7), further exemplifying the
challenging conditions of the DC-8 cabin for filter sampling.



The prior analysis made the assumption that the PM maintained a spherical shape upon
impacting the Teflon filter. More viscous (i.e., solid) PM is more likely to maintain a spherical
shape on filters whereas less viscous (i.e., liquid) PM will spread and become more similar to
cylindrical shape (e.g., Slade et al., 2019). As more acidic aerosol is more likely to be liquid
(e.g., Murray and Bertram, 2008), an exploration of cylindrical shape was conducted. Depending
on the assumed height of the cylindrical shape, the timescale for a molecule of ammonia to
interact with a molecule of sulfuric acid decreases from ~5 s (for maximum ammonia and
aerosol volume) to ~4 s (assuming height of cylinder equals radius of sphere) to less than 1 s as
height decreases from 25 nm or less. The aerosol deforming and spreading upon impacting the
filters increases the particle surface area, and decreases the amount of time for cabin ammonium
to interact with the acidic PM. Thus, less viscous aerosol has more rapid uptake and interaction
with ammonia due to the higher surface area.
A potential limitation to the model is the accommodation coefficient of ammonia to
acidic PM, as there are conflicting reports on its value  (Hanson and Kosciuch, 2004; Worsnop et
al., 2004). However, as shown in Worsnop et al. (2004), once the sulfuric acid weight percentage
is 50% or greater, the different studies converge to an accommodation coefficient of ~1. Various
studies indicate that the RH in the cabin of jet airplanes is low due to how air is brought into the
airplane, typically < 20% (Hunt and Space, 1994; Brundrett, 2001; National Research Council,
2002). Even though the ambient RH may be higher than the RH in the cabin of the DC-8, the
water equilibration is rapid (< 1 s) for the temperature of the cabin of the DC-8, even for very
viscous aerosol (Shiraiwa et al., 2011; Price et al., 2015; Ma et al., 2019), meaning the PM on the
filter would rapidly reach equilibrium with the cabin RH upon exposure. This would result in a ≥



60% sulfuric acid weight percentage (Wilson, 1921) for the typical RH ranges in the cabin of
typical airlines. However, various measurements in the DC-8 cabin indicate the RH is $\leq 40\%$
(Fig. S9), leading to sulfuric acid weight percentage of 50% or greater (Wilson, 1921).
Therefore, the accommodation coefficient of ~1 is well-constrained by the literature. Thus, the
handling of the filters between the sampling inlet to the Teflon bag exposes the acidic PM to
enough gas-phase ammonia towards forming ammonium bisulfate or ammonium sulfate, biasing
high ammonium from the filters. This explains the differences seen in Fig. 1 – Fig. 4.

Another potential limitation is that the PM on the filters could form a layer, as multiple

particles pile up on top of each other, slowing the diffusion of ammonia to be taken up by acidic
PM. The filters have a one-sided surface area of $6.4\times10^{-3}$ m$^2$, while an individual particle at the
mode of the volume distribution  (Fig. 7) has a projected surface area of ~$7.1\times10^{-14}$ m$^2$. Thus,
~$9.0\times10^{10}$ particles would need to be collected to form a single layer of PM on the filter. The
number of molecules in a single particle of the mode size is ~$1.4\times10^{8}$ molecules (Eq. S2).
Therefore, ~$1.3\times10^{19}$ molecules need to be collected onto the filters in order to form a monolayer
of PM, which is equivalent to ~$2.2\times10^{3}$ µg total aerosol collected or ~700 µg sm$^{-3}$ aerosol
concentration. As the mass concentration in ATom was typically ~1 µg sm$^{-3}$ (Hodzic et al.,
2020), and total aerosol concentrations that high is rarely seen except for extreme events (such as
the thickest fresh wildfire plumes), it is very unlikely that more particle layering would delay the
diffusion of ammonia to acidic PM.

Various sensitivity analyses of the uptake of ammonia to sulfuric acid were conducted.

First, there is minimal impact of cabin temperature on the results. Though there was a 25 K range
in cabin temperature (Fig. S2), the impact on the molecular speed of ammonia in the model (Eq.



S1) leads to a ±2% change in molecular speed, resulting in small changes in the time. Further,
only large changes in the accommodation coefficient with temperature occurs for sulfuric acid
weight percentages < 40% (Swartz et al., 1999), which is smaller than the weight percentage
expected for the filters in the cabin of the DC-8. For the temperature range of the cabin of the
DC-8 (Fig. S2), the coefficient changes by less than 10%, which leads to a total maximum
change in Fig. 7 of ±12%. The largest impact on the results in Fig. 7 is changing the
accommodation coefficient. Reducing the accommodation coefficient by a factor of 10, though
not representative of the DC-8 cabin conditions, would mean the acidic PM would need to be
exposed to ammonia for ≥ 1 minute (Fig. S10). It is expected that the lower accommodation
coefficient will occur for conditions with higher RH (>80%), suggesting typical laboratory
conditions (along with the lower ammonia mixing ratios) or ambient conditions may experience
lower ammonia uptake to acidic PM. Finally, organic coatings may impact the accommodation
coefficient of ammonia to sulfuric acid; however, the amount of reduction on the accommodation
coefficient has varied among studies (e.g., Daumer et al., 1992; Liggio et al., 2011). Daumer et
al. (1992) showed no impact; whereas, Liggio et al. (2011) found a similar impact to reducing the
accommodation coefficient by a factor of 10 (Fig. S10). Thus, the results in Fig. 7 are in line
with Daumer et al. (1992) while the results in Fig. S10 are in line with Liggio et al. (2011).

**3.4 Impacts of Ammonia Uptake on Acidic Filters**
As discussed throughout this study, uptake of cabin ammonia during the handling of
acidic filters can lead to biases in ammonium mass concentrations. However, other potential
sources of biases include the material used for sampling and storing the filter (Hayes et al.,



1980), and the preparation of the filter in the field to be sampled by ion chromatography. As the
preparation of the filters occurs indoors, as well, the filters will be exposed to similar ammonia
mixing ratios to those shown in Fig. 6.
Further, filter collection of aerosols is a widely used technique outside of aircraft
campaigns, including for regulatory purposes and long-term monitoring at various locations
around the world. For many of these sites, ammonia denuders are used to minimize biases of
ammonium on filters (e.g, (Baltensperger et al., 2003)). Data from remote, high altitude locations
have indicated that the ammonium balance is less than one (Cozic et al., 2008; Sun et al., 2009;
Freney et al., 2016; Zhou et al., 2019), similar to the observations from the AMS shown in Fig.
3. However, this is dependent on air mass origin and type (Cozic et al., 2008; Sun et al., 2009;
Fröhlich et al., 2015). Thus, sampling of remote aerosols with filters does provide evidence of
ammonium balances less than one due to a combination of procedures to minimize interaction of
gas-phase ammonia with the acidic filters and the lower human presence (and potentially cooler
temperatures at high, remote, mountaintop locations such as Jungfraujoch).
However, there are some long-term monitoring stations that do not use denuders or other
practices to minimize the interaction of ammonia with acidic aerosols. For example, the Clean
Air Status and Trends Network (CASTNET), which is located throughout the continental United
States, measures ammonium, sulfate, and nitrate (Solomon et al., 2014). The CASTNET system
uses an open-face system to collect aerosols on Teflon filters for approximately one week for
each filter (Lavery et al., 2009). In comparison, the Chemical Speciation Monitoring Network
(CSN), which also samples the continental United States and collects the aerosols on Nylon or
Teflon filters, a denuder is used to scrub gas-phase ammonia to minimize interaction of ammonia





with acidic aerosols on filters (Solomon et al., 2000, 2014). The comparison between these two
long-term monitoring sites show very different trends of ammonium balance versus total
inorganic mass concentration (Fig. S11). For CSN, the ammonium balance decreases with mass
concentration whereas CASTNET remains nearly constant. This is similar to the comparison
between SAGA and AMS in Fig. 4. This difference between the two sampling techniques may
be due to the lack of denuder in CASTNET to remove gas-phase ammonia. The use of the
denuders has led to CSN and other monitoring networks that use denuders to be more in-line
with in-situ observations (Kim et al., 2015; Weber et al., 2016). Further, as shown in Fig. S8,
exposure of an unprotected acidic filter for time greater than 1 day will lead to ammonia reacting
with the acid to form ammonium bisulfate or ammonium sulfate, even at low ammonia mixing
ratios. Thus, without denuders, or handling of filters with more than one person present, will lead
to similar differences between in-situ sampling versus filter collection of inorganic aerosols
observed during various aircraft campaigns.
Further, the uptake of ammonia onto acidic aerosols will impact comparisons with
chemical transport models (CTMs) and the understanding of various physical processes. For
example, various CTMs predict different results for the mass concentration of ammonium in the
upper troposphere (Wang et al., 2008a; Fisher et al., 2011; Ge et al., 2018), and selection of one
measurement versus the other will lead to different degrees of agreement. For example, for filters
that collect aerosols similar to those described here (no ammonia scrubber and/or exposed to
human emissions of ammonia), values of ammonium $< 0.2~\mu g~m^{-3}$ should not be used and either
disregarded or instead use *on-line* measurements of ammonium. This different agreement
impacts our understanding of important processes, such as the direct radiative impact of



inorganic aerosol (Wang et al., 2008b) or deposition of inorganic gases and aerosols (Nenes et
al., 2020a), as the gas-phase species have a faster deposition rate than the aerosol-phase. Finally,
the measurement biases can impact the suggested regulations to improve air quality (Nenes et al.,
2020b) and the calculated aerosol pH, as the pH is sensitive to the partitioning of ammonia
between the aerosol- and gas-phase (e.g., Hennigan et al., 2015).

**Conclusions**
Collection of aerosols onto filters to measure aerosol mass concentration and composition
is valuable for improving our understanding of the emissions and chemistry of inorganic aerosol,
and longstanding, multi-decadal filter-based records of atmospheric composition are invaluable
to analyze atmospheric change. However, as had been discussed in earlier studies, acidic aerosols
collected on filters are susceptible to uptake of gas-phase ammonia, which interacts with the
acidic aerosol to form an ammonium salt (e.g., ammonium bisulfate or ammonium sulfate). This
artifact in filter measurements can bias our understanding on the chemical composition of the
aerosol, which impacts numerous atmospheric processes.
We show that across six different aircraft campaigns, the aerosol collected on filters
showed a substantially higher ammonium mass concentration and ammonium balance compared
to AMS measurements. Further, another *on-line* measurement (PALMS) also shows lower
ammonium-to-sulfate ratios than for the filters. These differences are not due to differences in
the aerosol size ranges sampled by the PALMS and the filters. Instead, we show that the mixing
ratio of gas-phase ammonia in the cabin of the DC-8 is high enough (mean ~45 ppbv), and
similar to other indoor environments, to interact with acidic aerosol collected on filters in $\leq 10$ s,



to form ammonium salts. These results are consistent with prior studies investigating this

interference. Thus, due to the interaction of ammonia in the cabin of research aircraft, we suggest

that a more realistic limit-of-detection of ammonium is 200 ng sm$^{-3}$, versus the 10 ng sm$^{-3}$

typically cited based on the ion chromatography measurement. Finally, even though methods to

reduce this bias have been implemented in several ground-based long-term filter measurements

of inorganic aerosols, there are still some networks that collect inorganic aerosol without

denuders to remove gas-phase ammonia, leading to similar discrepancies between ground

networks as observed between filters and AMS on the various aircraft campaigns. Careful

practice in both the aerosol collection and filtering handling is necessary to better understand the

emissions, chemistry, and chemical and physical properties of inorganic aerosol.

**Acknowledgements**

This study was supported by NASA grants NNX15AH33A, NNX15AJ23G, 80NSSC18K0630, and 80NSSC19K0124. We thank Glenn Diskin and the DACOM team for the use of the CO$_2$ measurements from FIREX-AQ, Bruce Anderson, Luke Ziemba, and the LARGE team for the use of the LAS volume distribution from SEAC$^4$RS, Karl Froyd, Greggory Schill, and Daniel Murphy for the use of the PALMS observations from ATom-1 and -2, and Charles Brock, Agnieszka Kupc, and Christina Williamson for the volume distribution measurements during ATom-1 and -2. Also, we thank J. Andrew Neuman for the use of the Picarro G2103 during FIREX-AQ. We thank the crew of the DC-8 and C-130 aircraft for extensive support in the field deployments. We specifically thank Adam Webster and the crew of the NASA DC-8 in their assistance and persistence in allowing us to install the Picarro G2103 during FIREX-AQ in order to measure ammonia in the cabin.

**Data Availability**

ARCTAS-A and -B measurements are available at http://doi.org/10.5067/SUBORBITAL/ARCTAS2008/DATA001, last access 27 April 2020. SEAC$^4$RS measurements are available at http://doi.org/10.5067/Aircraft/SEAC4RS/Aerosol-TraceGas-Cloud, last access 27 April 2020. WINTER measurements are available at https://data.eol.ucar.edu/master_lists/generated/winter/, last access 27 April 2020. ATom-1 and -2 measurements are available at https://doi.org/10.3334/ORNLDAAC/1581, last access 27 April 2020. Ammonia and carbon dioxide measurements from the cabin of the DC-8 are available as an attachment . CSN and



CASTNET measurements are available at
http://views.cira.colostate.edu/fed/QueryWizard/Default.aspx, last access 27 April 2020.





**Figures**

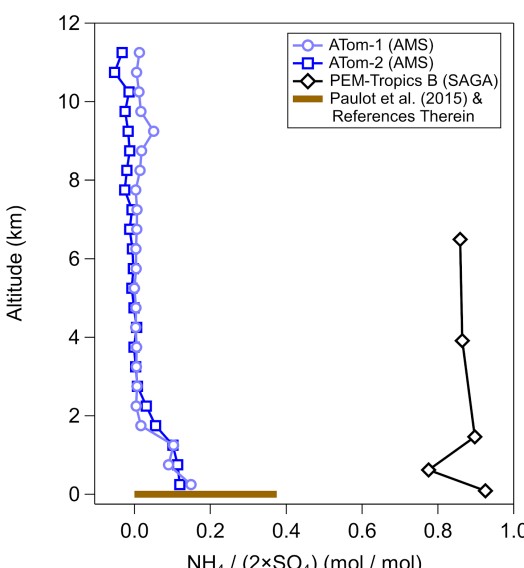

Figure 1. *Vertical profile of sulfate-only ion molar balance (moles($NH_4$)/moles($SO_4$)) measured during PEM-Tropics by collecting the aerosol on filters and analyzing it off-line with ion chromatography (Dibb et al., 2002) and during ATom-1 and -2 by AMS (Hodzic et al., 2020). The ammonium balance profile is for observations collected during ATom-1 and -2 between -20°S and 20°N in the Pacific basin, so that the observations were in a similar location as the PEM-Tropics samples. Also shown is the ammonium balance from observations summarized in Paulot et al. (2015), and reference therein, for the area around the same location as PEM-Tropics.*



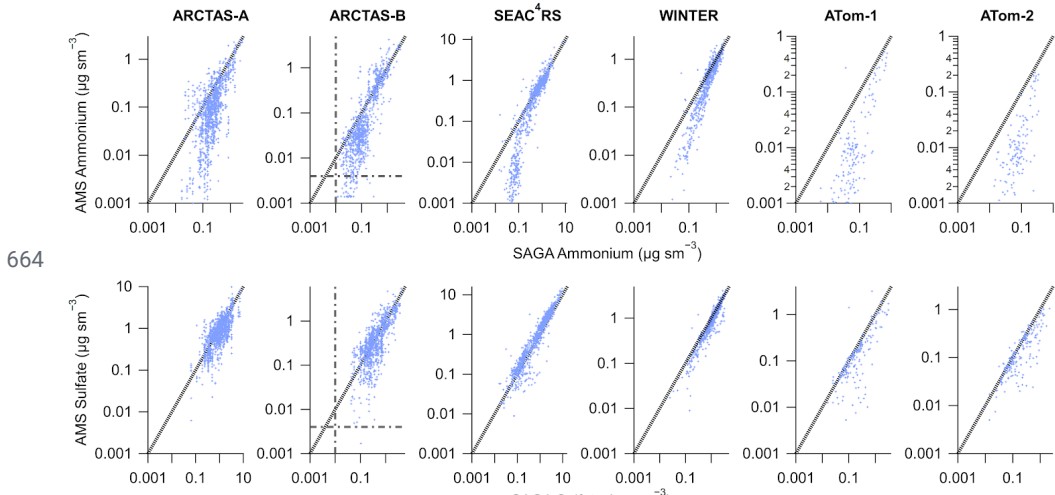

664

Figure 2. *Scatter plot of AMS (y-axis) versus SAGA filter (x-axis) ammonium (top) and sulfate (bottom) mass concentration from 6 different aircraft campaigns. AMS data have been averaged over the SAGA filter collection times. Black line is the one-to-one line and the grey dash-dot lines are the estimated detection limits for AMS (DeCarlo et al., 2006; Guo et al., 2020) at the SAGA filter collection interval (~5 minutes) and the estimated detection limits for SAGA (Dibb et al., 1999). Data has been averaged to the sampling time of SAGA and has not been filtered for supermicron particles. For ATom-1 and -2, data during ascent and descent has been removed (only level sampling at low altitude and high altitude).*

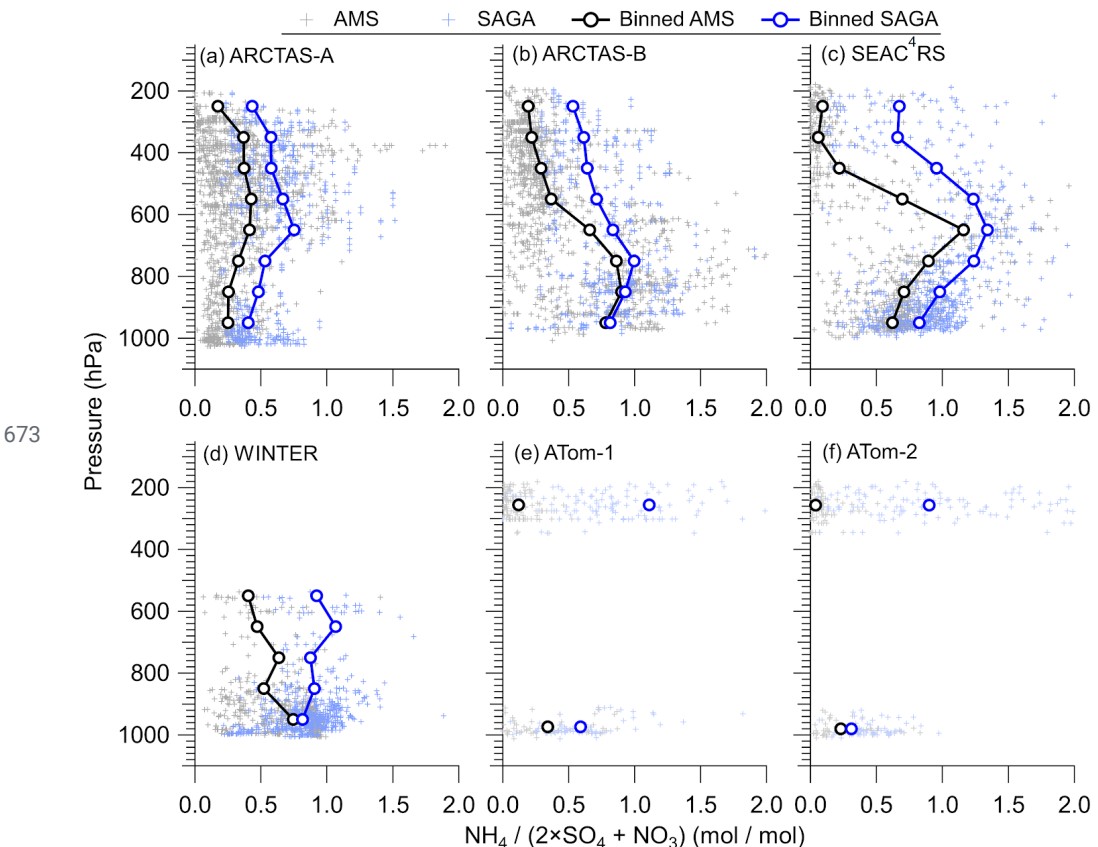

Figure 3. *Vertical profiles of ammonium balance (($NH_4$/18)/(2×$SO_4$/96+$NO_3$/62)) for (a) ARCTAS-A, (b) ARCTAS-B, (c) SEAC[4]RS, (d) WINTER, (e) ATom-1, and (f) ATom-2, for AMS and SAGA. The binned data is the mean for each 100 hPa pressure level. The data has been averaged to the sampling time of SAGA filters.*






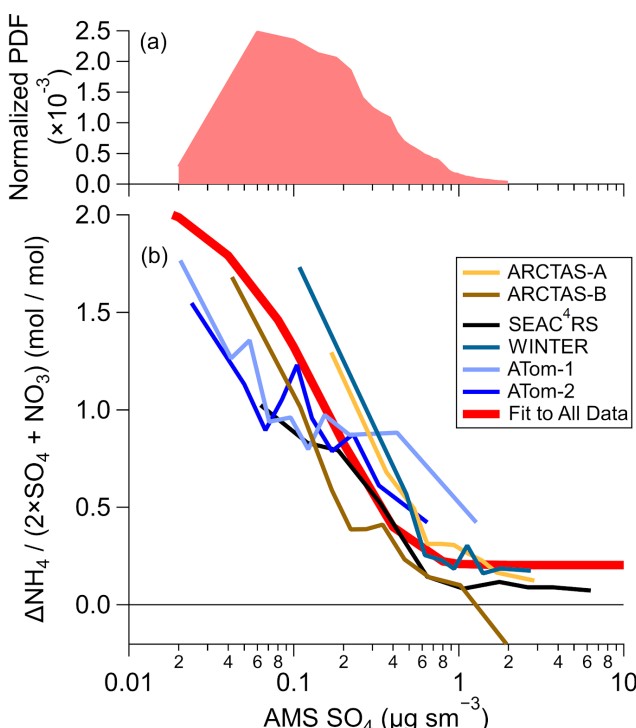

Figure 4. *(a) Predicted normalized probability distribution function (PDF) for tropospheric*
*(pressure > 250 hPa) sulfate from GEOS-Chem for one model year (see SI). (b) Difference*
*between SAGA and AMS ammonium, in mol sm⁻³, divided by AMS sulfate and nitrate, in mol*
*sm⁻³, versus AMS sulfate (μg sm⁻³), for the six different airborne campaigns. The values shown*
*are binned deciles for the five different airborne campaigns. The fit shown in (b) is for all data*
*from all campaigns.*

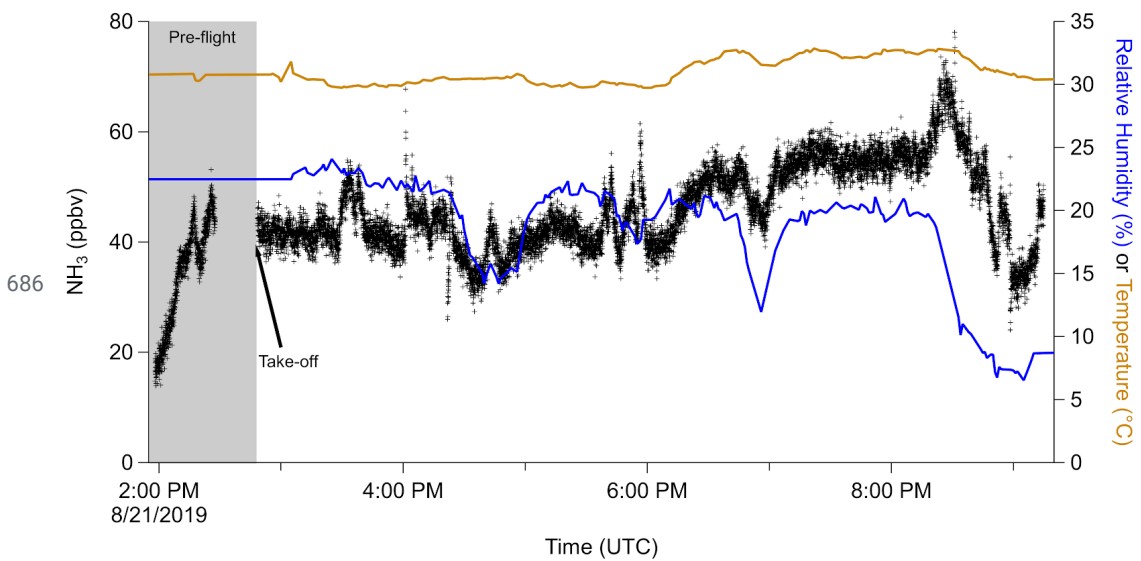


Figure 5. *Time series of ammonia (left) and relative humidity and temperature (right) measured*
*inside the cabin of the NASA DC-8 during a flight during the FIREX-AQ campaign. Time spent*
*prior to take-off is marked with a grey background.*



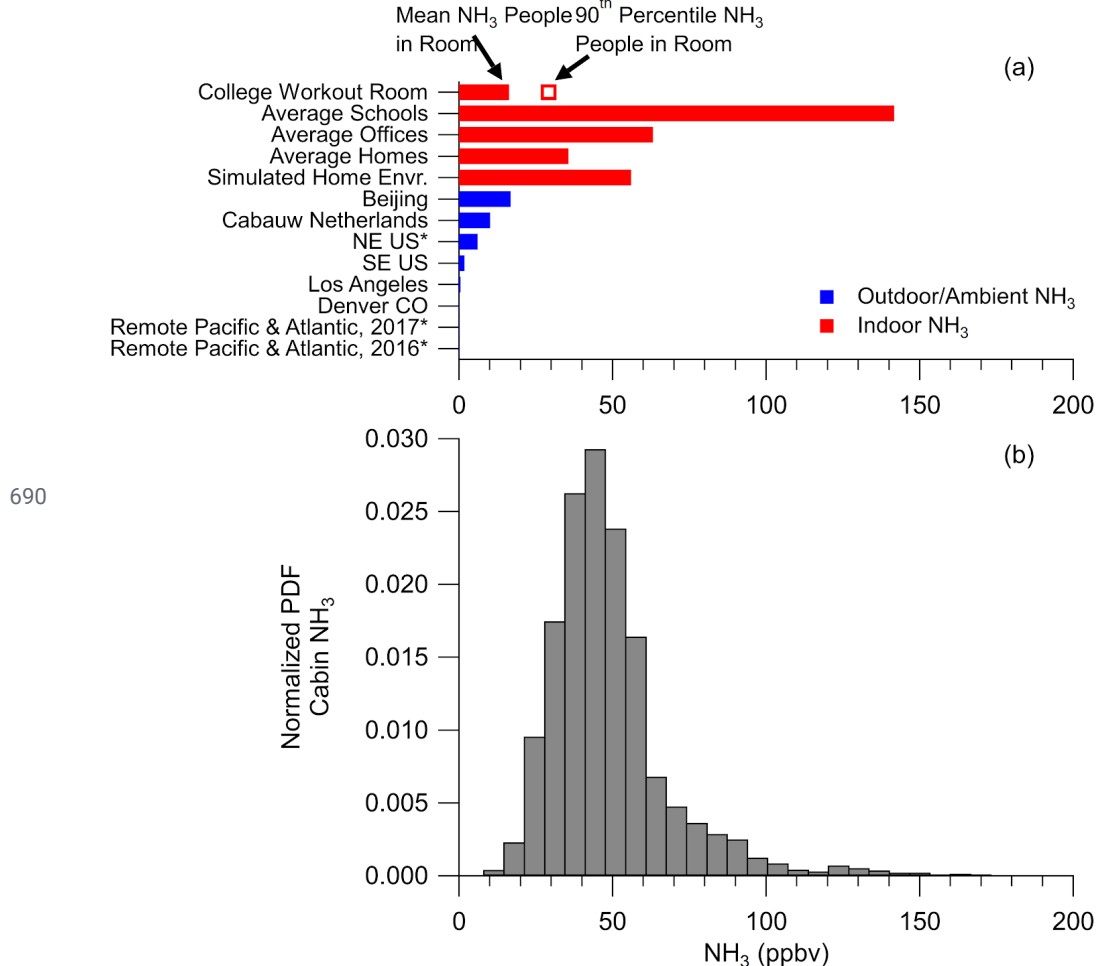


Figure 6. *(a) Ammonia (NH₃) (ppbv) reported for studies. See* Table S1 *for references. Asterisk after study name indicates NH₃ predicted by thermodynamic model instead of being measured. (b) Normalized probability distribution function (PDF) for NH₃, measured in the cabin of the NASA DC-8 during FIREX-AQ.*

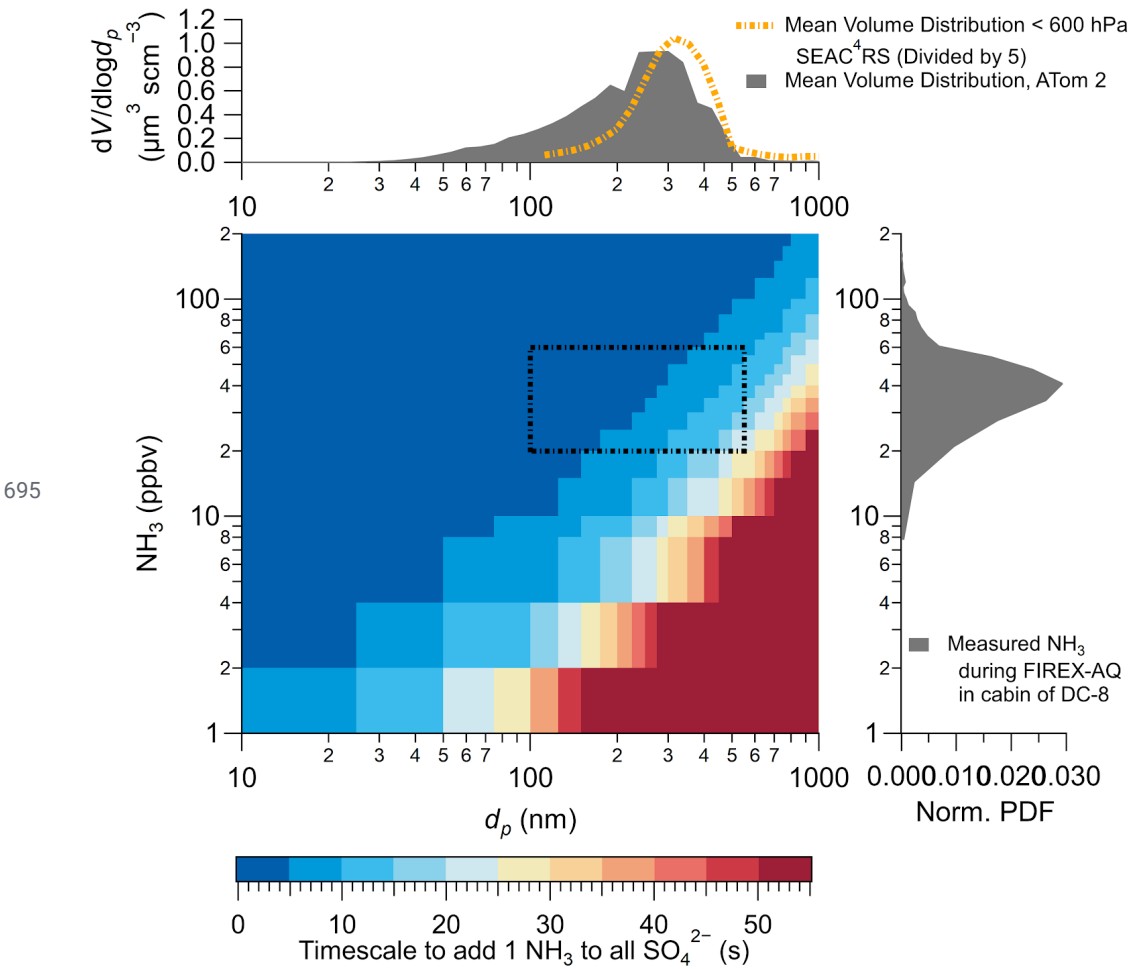

Figure 7. *Theoretical calculation for the amount of time it would take for all the sulfuric acid on the filter to react with one ammonia molecule to become ammonium bisulfate. Volume distribution is the average from SEAC⁴RS and ATom-2 (adapted from Guo et al. (2020)) and the normalized probability distribution function (Norm. PDF) is from* Fig. 6. *The representative diameter and ammonia mixing ratio are shown as dashed lines in the calculated timescale.*



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
