# Peer review of "Interferences on Aerosol Acidity Quantification due to Gas-phase Ammonia Uptake onto"

_Atmospheric Measurement Techniques, 2020_

## Referee Comment (RC1) · Anonymous Referee #1 · 11 Aug 2020

Nault et al. identify and characterize apparent artifacts associated with NH3 uptake onto acidic aerosols collected on filters during aircraft campaigns. This is important work and certainly relevant and useful for readers of AMT. It should be pointed out, however, that their results are not terribly surprising. It has long been recognized that filter samples of acidic aerosol need to be protected from human breath and other sources of ammonia during handling and storage. The fact that some practitioners of aerosol sampling by filters in aircraft campaigns (where acidic aerosols are even more likely to be encountered than at the typically more ammonia-rich surface) have ignored these lessons outlined in the literature is unfortunate.

[Figure]

I have several comments for the authors to consider in preparing a revised manuscript.

1. Abstract lines 45-47 and manuscript lines 376-379: the authors need to more fully specify the LOD they provide for filter sampling. This depends on a variety of factors, including sampled air volume and (depending on the stage where contamination occurs) filter extraction volume. At a minimum they should state their LOD estimate is appropriate for typical SAGA filter collection and extraction protocols.

2. Line 92: Please change "cations" to "anions." Sulfate and nitrate are anions.

3. Lines 178-181: Can the authors exclude loss of NH+ volatiles from the warming/drying of the AMS sample stream as a cause of some of the difference vs. filter NH4+ levels?

4. Lines 223-224: Plastics are common sources of NH4+ contamination vs. offgassing of NH3 adsorbed onto the plastic surface. Many researchers who are worried about artifact neutralization of acidity on aerosol filter samples use acid-coated substrates as NH3 sinks inside bags or other containers used for sample storage. Did the authors evaluate the polyethylene bags as a potential source of contamination? Were acid scrubbers inserted into the bags to prevent such an artifact from offgassed NH3?

5. Lines 225-226: I was shocked to see that collected filter samples were extracted with 20 mL of water. This represents a huge dilution when extracting a sample that has collected only 2-3 m3 of air. By diluting aqueous concentrations to low levels, any background NH4+ in the extract solution has an outsize effect on raising calculated aerosol ammonium concentrations and the uncertainty associated with measuring low extract ion concentrations is also magnified. Can the authors justify this large extraction volume and assess possible contributions to the concluded artifacts in the filter samples? A modern conventional IC analysis needs only 20-100 $\mu$L of injected volume (some capillary systems use far less) and even an autosampler can easily work with a total extract volume of several hundred $\mu$L.

6. Section 2.2.3. I am puzzled why the authors rely on PALMS data to get an independent (of AMS) estimate of online particle ammonium balance. The PALMS sulfate acidity indicator, as pointed out by the authors, is calibrated by comparison to PILS ion concentration ratios. The WINTER campaign flew with a PILS onboard. The authors should use that PILS ion balance directly rather than the PILS-calibrated PALMS data, which the authors point out can be influenced by changes in laser power. By its design and reliance on direct IC measurements of ion concentrations in aerosol extracts, the PILS should provide the most definitive measure of ratios of $NH_4^+$ to $SO_4^{2-}$.

7. Section 2.3.1. The FIREX campaign targeted smoke plumes. Biomass burning smoke can be very rich in $NH_3$. How much might penetration of smoky air into the aircraft cabin influenced the $NH_3$ concentrations measured there? The authors' air exchange measurements and calculated concentrations with assumed human $NH_3$ emission rates suggest that smoke $NH_3$ might not have been a major factor in determining cabin $NH_3$ concentrations. That surprised me!

8. Line 393: The filter storage bag here is specified to be Teflon vs. the polyethylene bag referred to earlier in the manuscript.

9. I like that the authors consider the timescale for diffusion to the collected aerosol particles in the filter. I do want to be sure they are calculating the timescale correctly. Can the authors please verify that the timescale expression they used (Eqn. 1) applies to a porous membrane? I am surprised that there is no dependence on pore size included. Also, what is the relevant timescale for $NH_3$ to diffuse into an acidic particle itself? It needs to do more than just reach the surface.

10. Pp. 26-27. The discussion of CSN and CASTNet $NH_4^+$ differences is interesting, although other factors beyond those discussed are likely at play. Both filter sampling systems can lose volatile $NH_4^+$ (e.g., $NH_4NO_3$). The degree of loss will increase in the denuded system as the equilibrium with the gas phase is strongly perturbed. Differences in sample handling, shipping, and storage may also be important.

11. Lines 589-592: The authors' computed 0.2 $\mu$g/m3 threshold is relevant for the SAGA system as used here but should not be more generally claimed for other filter-based sampling approaches with different sample volumes. Post-collection NH3 uptake will yield different impacts on ambient aerosol LODs in other systems.
* * *

---

## Referee Comment (RC2) · Anonymous Referee #2 · 12 Aug 2020

This manuscript provides a detailed analysis and discussion on artefacts related to filter handling and analysis during atmospheric measurements. For this discussion, the authors grouped together six different airborne measurement campaigns where both offline filters and online aerosol mass spectrometry were used to measure aerosol chemical composition. The authors highlight discrepancies in measurements that are thought to be largely related to handling artefacts and exposures of filters samples to ambient ammonia from the laboratory environment and from human interference. This work illustrates how artefacts related to sampling and handling of offline measurements can result in observations that can lead to the misinterpretation of atmospheric measurements, which will then inherently lead to discrepancies when comparing with global

transport models. The authors recommend that the limit of detection of ammonia on filters is increased and that when possible a denuder is used for filter sampling.

This manuscript is well written with well-illustrated figures and detailed supplementary information, and I recommend this manuscript for publication. I have a small number of remarks below related to additional information that could be included in the discussion.

Minor comments: Line 176: The AMS samples behind the NCAR inlet (HIMIL); the upper size cut of this inlet is not mentioned. (Line 216:The SAGA inlet is stated to have an aerodynamic diameter cut of 4.1 microns). Can the author include the upper size cut of the HIMIL inlet and that it was isokinetic sampling?

What was the flow rate of the SAGA inlet?

What is the lower size cut of these two inlets? Given that discrepancies between the two methods were highest as lowest mass concentrations, could they be a result of different sampling efficiencies for particles with diameters < 80 nm?

In section 2.2.2 Aerosol filters. There was no mention of filter blanks. Can the authors state how blank filter measurements were made (each flight or every couple of flights)?

There were several instruments operating together on the plane. Was a mass closure check performed on the AMS measurements to illustrate that this instrument was measuring all the NR-PM1? How did this mass closure change with altitude?

If measured, how did the OC/OM concentrations measured on the filters compare to the organic mass measured by the AMS instrument? Was the PILS instrument available on any of the flights? How did the PILS data compare with offline filters?

---

## Author Comment (AC1) · 25 Sep 2020

**Response to reviewers' comments on the paper "Interferences on Aerosol Acidity Quantification due to Gas-phase Ammonia Uptake onto Acidic Sulfate Filter Samples"**

We would like to thank both reviewers for their time and for their useful comments that have helped improve and clarify our paper. For ease, comments from reviewers are in black, responses in blue, and new text added to paper in **bold blue**.

*Reviewer #1*

1.0. Nault et al. identify and characterize apparent artifacts associated with NH3 uptake onto acidic aerosol collected on filters during aircraft campaigns. This is important work and certainly relevant and useful for readers of AMt. It should be pointed out, however, that their results are not terribly surprising. It has long been recognized that filter samples of acidic aerosol need to be protected from human breath and other sources of ammonia during handling and storage. The fact that some practitioners of aerosol sampling by filters in aircraft campaigns (where acidic aerosol are even more likely to be encountered than at the typically more ammonia-rich surface) have ignored these lessons outlined in the literature is unfortunate.

We agree that the importance of $NH_3$ uptake onto filters has been discussed in prior studies, which we already included in lines 84 through 87 of the AMTD version (Klockow et al., 1979; Hayes et al., 1980; Koutrakis et al., 1988). However, because the SAGA has been on many major airborne campaigns since the 1980s, the measurements from SAGA filters have been used to constrain chemical transport models (e.g., Wang et al., 2008a, 2008b; Ge et al., 2018), and the recent ATom campaigns that measured very remote air over Pacific, Atlantic, Southern, and Arctic Ocean. Thus, we felt it was important to analyze this uptake in regards to the SAGA system to ensure documentation as well as proper interpretation of past measurements in any future study, as we expect, e.g., the data collected from ATom to be used in numerous future studies. Finally, the very high speed of neutralization that we document is a novel and important aspect that has not been fully discussed in the prior studies we listed above.

I have several comments for the authors to consider in preparing a revised manuscript.

1.1. Abstract lines 45-47 and manuscript lines 376-379: the authors need to more fully specify the LOD they provide for filter sampling This depends on a variety of factors, including sampled air volume and (depending on the stage where contamination occurs) filter extraction volume. At a minimum they should state their LOD estimate is appropriate for typical SAGA filter collection and extraction protocols.

Here and throughout the rest of the manuscript, we have specified specifically for the SAGA system as flown on the DC-8 and have also noted that similar type analysis should be conducted for other filter systems but will lead to different results.

For line 47, we have changed it to say:

**"Finally, a more meaningful limit-of-detection for SAGA filters collected during airborne campaigns is ~0.2 μg sm$^{-3}$ ammonium, which is substantially higher than the limit-of-detection of the ion chromatography. A similar analysis should be conducted for filters that collect inorganic aerosol and do not have ammonia scrubbers and/or are handled in the presence of human ammonia emissions."**

For line 404, we have changed it to say:

**"Thus, this analysis suggests that for SAGA filters, a more meaningful ammonium limit-of-detection would be equivalent to 1 μg sm$^{-3}$ sulfate, which would be ~0.2 μg sm$^{-3}$ ammonium. This also provides the framework to define limit-of-detection for other filter-based measurements not associated with ion chromatography."**

For line 650, we have changed it to say:

**"For example, for filters that collect aerosols similar to those described here (no ammonia scrubber and/or exposed to human emissions of ammonia), values of ammonium < 0.2 μg sm$^{-3}$ should be used with caution or insead use *on-line* measurements of ammonium (specifically for SAGA measurements but a similar analysis should be conducted for other filter-based measurements)."**

Finally, for line 678, we have changed it to say:

**"Thus, due to the interaction of ammonia in the cabin of research aircraft, we suggest a more realistic limit-of-detection of ammonium for the SAGA filters is 200 ng sm$^{-3}$, versus the 10 ng sm$^{-3}$ typically cited based on the ion chromatography measurement."**

1.2. Line 92: Please change "cations" to "anions." Sulfate and nitrate are anions.

Changed.

1.3. Lines 178-181: Can the authors exclude loss of NH+ volatiles from the warming/drying of the AMS stream as a cause of some of the difference vs. filter NH4+ levels?

Yes. This has been analyzed in depth in prior studies (Guo et al., 2016, 2017; Shingler et al., 2016). For example, Guo et al. (2016) showed that for the residence time of the PILS inlet sample (~2 s) and the heating between ambient and cabin air (~17 K), the observed ammonium nitrate was inconsistent with a calculation that considered evaporation of ammonium nitrate. Instead, the observations were consistent with the calculation that assumed the ambient (277 K) vs the cabin (294 K) temperature. As the residence time for the AMS is faster than PILS (< 1 s) (Nault et al., 2018; Schroder et al., 2018; Guo et al., 2020), we do not expect any losses of these semivolatile compounds.

We have added the following lines, starting at line 195:

**"To minimize any potential losses of volatile aerosol components, the residence time between the inlet and AMS was less than 1 s (Nault et al., 2018; Schroder et al., 2018; Guo et al., 2020). Prior studies (Guo et al., 2016; Shingler et al., 2016) have shown minimal loss of semivolatile components for this residence time."**

1.4. Lines 223-224: Plastics are common sources of NH4+ contamination vs. offgassing of NH3 adsorbed onto the plastic surface. Many researchers who are worried about artifact neutralization of acidity on aerosol filter samples use acid-coated substrates as NH3 sinks inside bags or other containers used for sample storage. Did the authors evaluate the polyethylene bags as a potential source of contamination? Were acid scrubbers inserted into the bags to prevent such an artifact from offgassed NH3?

The following text has been added to SI (Sect. S4):

**"Research from co-authors on a prior paper showed that films of water are the most likely reason for the retention and slow release of sticky volatile gases from surfaces coated by Teflon and other surfaces. An upper limit water thickness is ~10 μm (Liu et al., 2019). The Henry's Law Coefficient for ammonia is 62 M atm$^{-1}$ (Seinfeld. and Pandis, 2006). With the bags being ~1.6×10$^4$ mm$^2$ (~1.6×10$^{-2}$ m$^2$), that would put an upper limit of water volume of ~1.6×10$^{-7}$ m$^3$ (~1.6×10$^{-4}$ L). The average ammonia in the cabin of the DC-8 was ~45 ppbv (~4.5×10$^{-9}$ atm), leading to ~2.8×10$^{-7}$ M ammonia partitioned to the water in the bag. Thus, that would lead to ~4.5×10$^{-11}$ mol ammonia on the walls, or ~2.7×10$^{13}$ molecules ammonia. The average number of sulfate molecules on the filters was ~3.8×10$^{15}$. Thus, at the upper limit for the water thickness of the bags, there is ~0.7% ammonia:sulfate molecules. As the bags are blown with dry air prior to placing the filters into the bags, the water thickness is expected to be lower (~0.1 μm), leading to a three order magnitude decrease for ammonia**

**molecules in the bag. Thus, the bags are not expected to be a large source of ammonia contamination. However, this effect has not been directly investigated experimentally."**

And the following text has been added to main text (Line 508) to reference SI:

**"Some studies have suggested that the bags used to store the filters may be a source of ammonia (e.g., Hayes et al., 1980); however, calculations indicate the bags would be a small source of ammonia (see Sect. S4)."**

Finally, the following text has been added to the paper to address the second point at Line 250:

**"No acid scrubbers were inserted into the bags to prevent any artifact from offgassing of ammonia."**

1.5. Lines 225-226: I was shocked to see that collected filter samples were extracted with 20 mL of water. This represented a huge dilution when extracting a sample that has collected only 2-3 m3 of air. By diluting aqueous concentrations to low levels, any background NH4+ in the extract solution has an outsize effect on raising calculated aerosol ammonium concentrations and the uncertainty associated with measuring low extract ion concentration is also magnified. Can the authors justify this large extraction volume and assess possible contributions to the concluded artifacts in the filter samples? A modern conventional IC analysis needs only 20-100 µL of injected volume (some capillary systems use far less) and even an autosampler can easily work with a total extract volume of several hundred µL.

The following has been added to SI (Sect. S2):

**"The 20 mL is thought to be a balance between a couple of competing factors. (1) The SAGA team wants to be confident that they are completely extracting the soluble material from the filters (recall, the filters are 90 mm in diameter). They had conducted testing when they first started operating on the NASA DC-8 (late 1980's-early 1990's) and established that this amount of water was necessary to fully extract the material. (2) To counter the dilution, the SAGA team uses a pre-concentrator column and large volume injections into the IC (~5 mL). These two aspects compensate for the greater dilution. (3) Finally, 5 mL is injected for both anions and cations (total 10 mL), and enough sample is left to conduct a follow-up injection if there was any concern about the data."**

1.6. Section 2.2.3. I am puzzled why the authors rely on PALMS data to get an independent (of AMS) estimate of online particle ammonium balance. The PALMS sulfate acidity indicator, as pointed out by the authors, is calibrated by comparison to PILS ion concentration ratios. The

WINTER campaign flew with a PILS onboard. The authors should use that PILS ion balance directly rather than the PILS-calibrated PALMS data, which the authors point out can be influenced by changes in laser power. By its design and reliance on direct IC measurement of ion concentrations in aerosol extracts, the PILS should provide the most definitive measure of ratios of NH4+ to SO42-.

For ATom, PALMS is the only other instrument that offers information on submicron aerosol acidity. Even if somewhat indirect, it is still useful, since the remote atmosphere is where the largest differences appear. PILS is not available except for WINTER and ARCTAS. Unfortunately, the PILS data from WINTER could not be used for this analysis. As discussed in Guo et al. (2016), the cation IC exhibited higher baseline noise during the WINTER campaign compared to the anion IC, leading to insufficient sensitivity for reliable ammonium measurements. Further, Schroder et al. (2018) found that the PILS sulfate mass concentration was lower than the AMS sulfate concentration (slope of AMS vs. PILS = 1.5, $R^2$ = 0.75), even though there was good agreement with the AMS and SAGA filter sulfate mass concentration (slope of AMS vs. SAGA = 1.0, $R^2$ = 0.92). Similarly, as shown below, the PILS sulfate was lower than the SAGA sulfate during WINTER. Thus, these factors make comparing against PILS during WINTER unreliable. Finally, a similar disagreement between PILS and SAGA was observed during ARCTAS campaigns, the only other campaign that PILS, SAGA, and AMS were co-located on the same plane, whereas SAGA and AMS showed similar agreement as WINTER (Aknan, 2015).

[Figure]

Figure 1. Scatter plot of PILS and SAGA sulfate during WINTER campaign.

We have added the following at line 384:

**"The only useful comparison, other than SAGA versus AMS, is with PALMS during ATom."**

1.7. Section 2.3.1. The FIREX campaign targeted smoke plumes. Biomass burning smoke can be very rich in NH3. How much might penetration of smoky air into aircraft cabin influenced the NH3 concentrations measured here? The authors' air exchange measurements and calculated concentrations with assumed human emission rates suggest that smoke NH3 might not have been a major factor in determining cabin NH3 concentrations. That surprised me!

We have conducted further investigation and added the following figure to the SI:

[Figure]

**Figure S7. (top) Average ambient ammonia, measured by PTR-MS (Müller et al., 2014), sampled in air influenced (HCN > 300 pptv) and not influenced (HCN < 300 pptv) by biomass burning during the time period cabin was being sampled by Picarro. Note, this sampling was weighted towards the time period that the DC-8 was sampling agricultural fires, where the plumes were significantly smaller (seconds) versus the western fires at the**

**beginning of the campaign (minutes - hours). (b) Normalized probability density function (PDF) of gas-phase ammonia (NH$_3$) measured in the cabin of the DC-8 during FIREX-AQ for when the DC-8 was sampling air influenced by biomass burning (HCN > 300 pptv) and not influenced by biomass burning (HCN < 300 pptv).**

And the following lines, starting at line 472:

**"During FIREX-AQ, the DC-8 frequently sampled air impacted by biomass burning, which is an important source of ammonia (Sutton et al., 2013) and could potentially increase the background ammonia being brought into and mixing with the cabin air being sampled by the Picarro. Splitting the cabin ammonia ratios between sampling air impacted by biomass burning versus nominally background air, the normalized PDF did not shift to higher ammonia mixing ratios (Fig. S7). Further, the averages of the observed cabin ammonia was statistically similar, at the 95% confidence interval, between the DC-8 sampling biomass burning and nominally background air (48.1±13.4 versus 44.1±14.4 ppbv for biomass burning and background air, respectively). Finally, the majority of the time the cabin air was sampled by the Picarro for cabin ammonia, the DC-8 was sampling agricultural fires in Southeast US, which are shorter in duration (seconds) versus the large wildfires in Western US (minutes to hours). This is reflected in the low average ambient value for ammonia, as measured by a proton transfer reaction mass spectrometer (Müller et al., 2014), when the DC-8 was sampling biomass burning-influenced air observed during this time (~10 ppbv) and very low average value for non-biomass burning-influenced air (~0.8 ppbv) (Fig. S7). Thus, ammonia from biomass burning would at most be a small impact on the ammonia measured in the cabin of the DC-8, further indicating the ammonia in the cabin was mainly from human emissions."**

1.8. Line 393: The filter storage bag here is specified to be Teflon vs. the polyethylene bag referred to earlier in the manuscript.

Corrected. The correct bag material is polyethylene bag.

1.9. I like that the authors consider the timescale for diffusion to the collected aerosol particles in the filter. I do want to be sure they are calculating the timescale correctly. Can the authors please verify that the timescale expression they used (Eqn. 1) applies to a porous membrane? I am surprised that there is no dependence on pore size included. Also, what is the relevant timescale for NH3 to diffuse into acidic particle itself? It needs to do more than just reach the surface.

The following has been added at line 518:

**"Even though the filters have a porous membrane, for molecular diffusion, the membrane only increases the pathway that the ammonia molecules have to travel slightly; thus, not changing the estimated time. Second, as the particles are liquid (Wilson, 1921), the diffusion will be similar as through water. A typical value for diffusivity in water is ~$1\times10^{-5}$ cm$^2$ s$^{-1}$ (Seinfeld and Pandis, 2006). For the size ranges observed (Fig. 7, ~40 - 700 nm), this corresponds to a timescale of $1.6\times10^{-7}$ to $5.0\times10^{-5}$ s. Thus, the diffusion through the filter and through the PM is nearly instantaneous for ammonia."**

1.10. Pp. 26-27. The discussion of CSN and CASTNet NH4+ differences is interesting, although other factors beyond those discussed are likely at play. Both filter sampling systems can lose volatile NH4+ (e.g., NH4NO3). The degree of loss will increase in the denuded system as the equilibrium with the gas phase is strongly perturbed. Difference in sample handling, shipping, and storage may also be important.

The following has been added at line 638:

**"Other aspects that could impact this comparison, and are beyond the scope of this study (but that have been discussed in other studies (Hering and Cass, 1999; Schauer et al., 2003; Chow et al., 2005, 2010; Dzepina et al., 2007; Watson et al., 2009; Nie et al., 2010; Liu et al., 2014, 2015; Cheng and He, 2015; Heim et al., 2020) include the loss of volatile ammonium from the evaporation of ammonium nitrate or differences in the handling, shipping, and/or storage of the filters or extracted samples."**

1.11. Lines 589-592: The authors' computed 0.2 μg/m3 threshold is relevant for the SAGA system as used here but should not be more generally claimed for other filter-based sampling approaches with different sample volumes. Post-collection NH3 uptake will yield different impacts on aerosol LODs in other systems.

Please see response to 1.1 above.

*Reviewer #2*

This manuscript provides a detailed analysis and discussion on artefacts related to filter handling and analysis during atmospheric measurements. For this discussion, the authors grouped together six different airborne measurement campaigns where both offline filters and online aerosol mass spectrometery were used to measure aerosol chemical composition. The authors highlight discrepancies in measurements that are thought to be largely related to handling artefacts and exposures of filters samples to ambient ammonia from the laboratory environment and from human interference. This work illustrates how artefacts related to sampling and handling of

offline measurements can result in observations that can lead to the misinterpretation of atmospheric measurements, which will then inherently lead to discrepancies when comparing with global transport models. The authors recommend that the limit of detection of ammonia on filters is increased and that when possible a denuder is used for filter sampling.

This manuscript is well written with well-illustrated figures and detailed supplementary information, and I recommend this manuscript for publication. I have a small number of remarks below related to additional information that could be included in the discussion.

Minor comments:

2.1 Line 176:  The AMS samples behind the NCAR inlet (HIMIL); the upper size cut of this inlet is not mentioned. (Line 216: The SAGA inlet is stated to have an aerodynamic diameter cut of 4.1 microns). Can the author include the upper size cut of the HIMIL inlet and that it was isokinetic sampling?

We have added the following lines, at line 183:

**"The best estimated upper size cut-off for the HIMIL inlet is ~1 μm diameter (geometric, David Rogers, pers. comm. 2011). This diameter is larger than the size cut-off than that of the AMS inlet (~0.5-0.7 μm diameter, geometric, depending on the composition), with no losses in the tubing between the HIMIL and AMS inlet expected (see Guo et al. (2020) for more details). Multiple comparisons with instruments sampling from an isokinetic inlet $PM_4$ inlet (Brock et al., 2019; Guo et al., 2020) indicate that no significant sampling biases were incurred over the size range of the AMS."**

2.2 What was the flow rate of the SAGA inlet?

We have added the following lines, at line 245:

**"The aerosol inlet flow is close to 400 slpm in the marine boundary layer and approximately 150 slpm at maximum altitude."**

2.3 What is the lower size cut of these two inlets? Given that the discrepancies between the two methods were highest as  lowest mass concentrations, could they be a result of different sampling efficiencies for particles with diameters < 80 nm?

The following has been added to line 238:

**"The lower size cut-offs for SAGA and AMS are similar (Guo et al., 2020). As discussed by Guo et al. (2020; their Fig. 8) the difference in mass sampled at the smaller sizes between SAGA and AMS is generally negligible at all altitudes."**

2.4 In section 2.2.2 Aerosol filters. There was no mention of filter blanks. Can the authors state how blank filter measurements were made (each flight or every couple of flights)?

The following has been added to line 247:

**"Further, 2 blank filters are collected each flight."**

2.5 There were several instruments operating together on the plane. Was mass closure check performed on the AMS measurements to illustrate that this instrument was measuring all the NR-PM1? How did this mass closure change with altitude?

We have added the following lines, at line 215:

**"Mass and/or volume closure has been investigated between the AMS and other measurements for all campaigns discussed here (Cubison et al., 2011; Aknan, 2015; Liu et al., 2017; Nault et al., 2018; Schroder et al., 2018; Guo et al., 2020). The closure was complete for the size range of the AMS and did not show any dependence with altitude (Guo et al., 2020)."**

2.6 If measured, how did the OC/OM concentrations measured on the filters compare to the organic mass measured by the AMS instrument? Was the PILS instrument available on any of the flights? How did the PILS data compare with offline filters?

The only organic molecule reported from SAGA filters is oxalate (Talbot et al., 1992; Dibb et al., 1997). Thus, a comparison of OC/OM between filters and AMS cannot be conducted.

Please see comment 1.6 above concerning PILS.

**References**

[revised manuscript text omitted]